# From cognitive control to visual incongruity: Conflict detection in surrealistic images

**Manuela Ruzzoli**[1,2]\*, **Aoife McGuinness**[1], **Luis Morís Fernández**[1,3], **Salvador Soto-Faraco**[1,4]

**1** Multisensory Research Group, Center for Brain and Cognition, Universitat Pompeu Fabra, Barcelona, Spain, **2** Centre for Cognitive Neuroimaging, Institute of Neuroscience and Psychology, University of Glasgow, Glasgow, United Kingdom, **3** Departamento de Psicología Básica, Universidad Autónoma de Madrid, Madrid, Spain, **4** Institució Catalana de Recerca i Estudis Avançats (ICREA), Barcelona, Spain

\* manuela.ruzzoli@gmail.com

**Data Availability Statement:** Data are available at: https://osf.io/wftd7/, together with the pre-registration document. Shared data includes the raw EEG data, the list of electrodes, the memory test data for all the participants included in the

## Abstract

This study explored brain responses to images that exploit incongruity as a creative technique, often used in advertising (i.e., surrealistic images). We hypothesized that these images would reveal responses akin to cognitive conflict resulting from incongruent trials in typical laboratory tasks (i.e., Stroop Task). Indeed, in many surrealistic images, common visual elements are juxtaposed to create un-ordinary associations with semantically conflicting representations. We expected that these images engage the conflict processing network that has been described in cognitive neuroscience theories. We addressed this hypothesis by measuring the power of mid-frontal Theta oscillations using EEG while participants watched images through a social media-like interface. Incongruent images, compared to controls, produced a significant Theta power increase, as predicted from the cognitive conflict theory. We also found increased memory for incongruent images one week after exposure, compared to the controls. These findings provide evidence for the incongruent images to effectively engage the viewer's cognitive control and boost memorability. The results of this study provide validation of cognitive theories in real-life scenarios (i.e., surrealistic ads or art) and offer insights regarding the use of neural correlates as effectiveness metrics in advertising.

## Introduction

Surrealistic techniques, an outgrowth of the early 20th-century artistic movement, make a point of deliberately defying reason. Surrealistic techniques have had a profound influence in film, visual arts and in advertising, where they have become a form of communication strategy [1,2]. Often, these techniques consist of the juxtaposition of common elements with semantically incongruent representations to create un-ordinary meanings. Here, we address the hypothesis that the incongruity generated by the juxtaposition of visual elements like in surrealistic techniques, engages a brain network involved in other forms of conflict situations, such as cognitive conflict.

analysis as presented in the manuscript and a file with specifications to read the data.

**Funding:** This research was supported by the Ministerio de Economía y Competitividad (PSI2016-75558-P AEI/FEDER), AGAUR Generalitat de Catalunya (2017 SGR 1545), and the European Research Council (PoC- 727595 to SSF) and. M.R. was supported by a Marie Skłodowska-Curie fellowship (Ctrl Code – 794649 - H2020-MSCA-IF-2017 to MR).

**Competing interests:** The authors declare no competing financial interests.

Cognitive conflict is defined as "the simultaneous activation of incompatible and competing representations" [3]. It is typically tested in well-controlled laboratory protocols such as the Stroop task [4], the Simon task [5] or the Flanker task [6], which notably produce consistent behavioral [7] and physiological outcomes [8–10]. In these tasks, mismatching information between stimuli features or response options (i.e., potential sources of cognitive conflict) leads to slower and more error-prone responses. Cognitive conflict is associated with increases in BOLD responses in the anterior cingulate cortex—ACC (measured by fMRI) [8,9] and in Theta oscillatory activity (4–8 Hz) over mid-frontal sensors (measured by EEG or MEG) [10,11].

The conflict monitoring theory [12–15] proposes that the ACC is responsive to the presence of conflict in information processing, and suggests that the function of the ACC is to trigger adjustments in cognitive control to flexibly allocate resources in order to minimize or resolve the conflict, for example via attention [3,12].

In the present study, we tested the hypothesis that images that juxtapose incongruent meanings, such as ads exploiting surrealistic techniques, will produce conflict adjustments like those in classical conflict protocols. Namely, an increment in frontal-medial Theta oscillatory power (4-8Hz), compared to control images that do not contain incongruity. To the best of our knowledge, only one study has tested the brain responses to surrealistic images in advertising by functional magnetic resonance (fMRI) and showed, amongst other areas, greater activation in the anterior cingulate cortex (ACC) compared to controls [16]. Mostafa [16] interpreted the ACC activation in association with the detection of novel elements (or violation of expectations) in surrealistic images compared to controls.

Furthermore, in the present study, we assessed the impact of conflict on subsequent memory by introducing a recognition task one week after the first exposure to both the incongruent and control images during the EEG test. Visual memory capacity has been linked to semantic (in)consistency [see 17], whereby objects that are inconsistent with the surrounding scene are remembered more [see also 18,19]. However, the impact of surrealistic ads on memory has been only tested in a seminal study by Homer & Kahle (1986) using a free-recall test. They found that participants who saw surrealistic ads were less inclined to errors, compared to a control group, thus supporting the proposal that surrealistic images strengthen memory. Although the advertising industry often exploits conflict to boost memorability, our study seems to be the first that addresses this assumption from a neuroscientific perspective, combining electrophysiological and behavioral responses in a hypothesis-driven manner.

## Methods

### Pre-registered protocol

The initial hypotheses, procedure and analytical pipeline were pre-registered in the OSF platform (https://osf.io/xa7rv/) before data collection. Here we present the results obtained in the pre-registered analyses, and further exploratory analyses carried out as follow-ups (specified below).

### Participants

We recorded data from 31 healthy participants (15 females) aged between 18–34 years (22 ± 3), with normal or corrected-to-normal vision. Data from two participants were excluded because of a technical problem in EEG data storing (N = 1), and because of a failure to comply with the instructions in the memory task (N = 1). After EEG artefact rejection, data from two further participants were excluded because less than 30 artefact-free trials were left for analysis. A total of 27 participants were thus included in the analyses. Only for the Theta

peak analysis, data from one further participant were excluded because no peak in the Theta range could be found in the control condition, leaving a final sample of 26 participants for that particular analysis. The experiment was run following the Declaration of Helsinki and approved by the ethics committee CIEC Parc de Mar (Universitat Pompeu Fabra, Barcelona, Spain). Participants gave written informed consent before participating in the study and received a compensation of 10 euro/hour for their time.

## Apparatus & stimuli

The experiment was designed and performed using Psychtoolbox [20] on Matlab R2016b. Visual stimuli were presented through a CRT monitor (1024 x 768 pixels), with a refresh rate of 60Hz and 32-bit color resolution. A set of 120 incongruent images were selected from image directories such as Pinterest and Google Image Search, with the following search queries: 'Creative Advertising', 'Smart Ads', 'Innovative Advertising', and 'Surrealist Ads'. One hundred and five of the selected images were advertisements, and 15 were similar artistic images. The selection of the set of incongruent images was initially based on the author's subjective evaluation of whether it could be considered 'out of the ordinary' in terms of the semantics or visual content and whether it contained two or more interacting objects that are usually semantically unrelated (see Fig 1A for an example).

One control image was selected for each of the 120 incongruent images based on conceptual similarity (e.g., content, main element in the scene, background) as well as visual similarity (e.g., colors, light, camera angle). Although it is impossible to perfectly match all low-level features between our realistic images (incongruent images and controls), we used 'Google Reverse Image Search' and the website 'Yandex' (https://yandex.com/), to control for this limitation and objectively define visually similar images. The reverse image search is a content-based image retrieval (CBIR) technique that searched for another image whose content, dominant

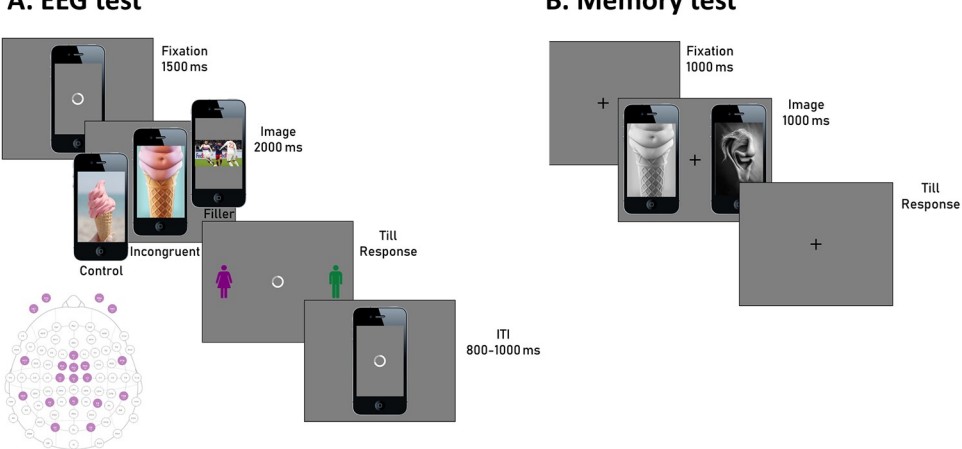

**Fig 1. A**. Schematic representation of a trial during the EEG recording session. A trial started with a 1500 ms fixation (loading wheel), followed by the target image (incongruent, control or filler) presented for 2000 ms. A response screen prompted the participants to choose whether they thought the presented image was directed to females or males (the position of the icons was chosen randomly at each trial) and lasted till the response was provided. The next trial started after a variable inter-trial interval (ITI) randomly jittered between 800 and 1000 ms. A representation of the 16 EEG electrodes montage is also provided. **B**. Schematic representation of the memory test performed about one week after the EEG test. Participants were asked to fixate on a fixation presented for 1000 ms, before the image pair appearance. They had to choose which of the presented images (both incongruent and control) had been presented the week before during the EEG test.

color and other features are like the image provided to the system (i.e., incongruent image). All images were edited in Photoshop (version CS6, 2012) to remove text and any branding (e.g., logos) if present.

All 240 images initially selected with the methods above were then subjected to a survey to validate the levels of visual incongruity for each image by independent judges. To do so, we designed a survey based on a prior study on creativity and divergence by Smith et al. [21]. We selected the items regarding the categories of *originality* and *synthesis*, out of the seven defined in the original study. Although Smith et al. [21] broadly investigate the determinants of creativity in ads, we only focused on the possibility to objectively determine image incongruity. We asked the participants to score each image from 1 (total disagreement) to 7 (full agreement) according to the following statements:

1. The image is out of the ordinary.

2. The image connects objects that are usually unrelated.

To shorten the duration of the survey for each participant, we divided the whole set of 240 images into three equivalent 80-image surveys randomly distributed online via the Qualtrics survey platform [22]. We obtained data from a total of 122 respondents (N = 48, 38, and 36, for Surveys 1, 2, and 3, respectively), who were not involved in the subsequent tests.

The results of the survey were then used to select images for the experiment. Specifically, as incongruent images, we selected images with a mean rate of 4 and above (incongruent images, with a high level of agreement with the two statements in the survey). Images rated less than 4 were selected as control images (lowest level of agreement with the two statements in the survey). The top 100 images in each condition were then selected (200 in total) as the experimental image set. Incongruent images had an average rating of 5.58 ± 1.33 for the "originality" domain (question 1 in the survey) and 4.26 ± 1.79 for the "synthesis" domain (question 2 in the survey). Control images had an average rating of 2.09 ± 1.24 and 1.87 ± 1.09, respectively. It is important to keep in mind that the categories of originality (the image is out of the ordinary) and synthesis (the image connects objects that are usually unrelated), used here as instruments to select images for the incongruent and control sets, are individual items that were validated in a wider study on divergent thinking in advertising [21]. Other instances of incongruency (e.g., novelty, divergence, flexibility) have not been considered in the present image sets, and therefore these items alone may not be representative of the whole construct defined in the original work.

The experiment also included filler images based on trending topics on Twitter (circa March 2018) in various locations worldwide. A random mix of images related to current affairs, pop culture, sports and entertainment were chosen to mimic social media news feeds. Filler images were not analyzed and were not included in the image selection survey. We decided to include fillers only because we wanted to create a realistic social media context. In addition, we assumed that interspersed filler images would dilute the contrast between incongruent and control stimuli and, would, by increasing the image set, make the subsequent memory task more difficult. A total of 250 images (100 incongruent + 100 controls + 50 fillers) were used for the EEG experiment.

For the test during the EEG recording, the 100 incongruent images and their corresponding 100 controls were split into two equivalent lists (counterbalanced across participants) so that only one image from each pair (either the incongruent or the control) appeared to each participant, but all images appeared the same number of times across the whole experiment. In addition to the 50 control and 50 incongruent images, each list also included all the 50 fillers. Each participant viewed only one image list during the EEG recording and was tested using all images (from both sets) in the subsequent memory test.

## EEG recording

During the exposure session, EEG was continuously recorded through 16 electrodes using a portable EEG system (Enobio, Neuroelectrics, Barcelona, Spain) with a custom electrode montage, focused around the mid-frontal area (see Fig 1). Horizontal and vertical electro-oculograms (hEOG and vEOG) were recorded from two additional electrodes placed at the outer canthus and under the right eye, respectively. Two additional electrodes were placed over the left and right mastoids for off-line re-referencing. Online reference electrode and ground were clipped onto the left earlobe.

In the spirit of the ecological validity underlying this study, we attempted to test our hypotheses under close to realistic conditions to improve the feasibility of applying such measurements outside the laboratory [23,24]. Therefore, we used a wearable EEG headset that makes it more agile and applicable to real-world testing. This 16-electrode montage proved enough to measure mid-frontal Theta power (Fig 1). The theory-driven approach adopted here helped to narrow down on specific scalp sites and compensate for the relatively sparse spatial sampling.

## Experiment 1: EEG test

The EEG experiment was performed in a dimly lit room and lasted approximately 1 hour. Each participant viewed 150 images (50 incongruent, 50 control and 50 filler) in random order. Participants were asked to decide whether they thought the image was directed to a male or female audience, to proceed to the next image. This task was orthogonal to the goals of the study and was devised merely to ensure that participants looked at each image closely during the EEG phase. No specific analyses were conducted on this response.

Participants rested their hands palm down, with the index fingers positioned on the response keys of the computer keyboard ('z' and 'm' for left- and right-hand responses, respectively). At the beginning of each trial, a frontal view of a smartphone displaying the typical loading wheel on the screen (mimicking a social media load screen, where participants would browse images amongst other information related to current news, in a self-paced manner) was presented for 1500 ms, which served a as fixation point. Next, a randomly chosen incongruent, control or filler image appeared on the smartphone screen. After 2000 ms, a male and female icon appeared on either side of the image, signaling to participants that they could respond using the 'z' and 'm' keys, corresponding to a gender. The position of the male and female icons was randomized across trials so that participants could not prepare their response in advance, avoiding motor response contamination in the EEG signal. The following trial started after a randomly jittered period (800 to 1000 ms) following response. Each participant ran a total of 5 experimental blocks of 30 trials each.

## Experiment 2: Memory test

The primary aim for the memory test was to address whether incongruent images were spontaneously remembered better than controls. Also, overall performance in the memory test would work as a reality check, ensuring that participants looked at the images during the EEG experiment. We invited participants to come back to the lab one week after the exposure phase with EEG recordings and to run a 2AFC (two-alternative forced-choice) task. Before or during the EEG experiment, participants were not informed that they would be asked to perform a memory test one week later. In the memory task, participants had to choose which of two images presented side-by-side they remembered seeing during the study session, one week before. The memory test comprised of 200 images (100 pairs) in total. That is, the full sets of congruent and incongruent images, including the 50 incongruent (and 50 congruent) images

previously seen, randomly paired with the 50 incongruent (and 50 congruent) images which had not been seen during the EEG recording. All images were transformed into black and white to encourage participants to rely on their conceptual memory of images, rather than perceptual features [25]. Participants rested their hands palm down, with the index fingers positioned on the response keys of the computer keyboard ('z' and 'm' for left- and right-hand responses, respectively). A fixation cross appeared in the middle of the screen, along with the image pairs (1000 ms) presented laterally. Participants were instructed to respond as fast as possible, indicating which one of the two presented images they remembered having seen during the EEG experiment. Once the images disappeared, the fixation-cross remained until participants pressed either the 'z' or 'm' key, corresponding to the image on the left and the right-hand side of the fixation-cross. The position of the seen image in the pair (correct response) was counter-balanced. The next pair of images appeared 1000 ms after the response (Fig 1B).

## Data pre-processing

EEG data was high-pass filtered from 0.5 and low-pass from 40 Hz (Butterworth filters of order 2 and 8 respectively), with an additional notch filter at 50 Hz. Although our initial intention was to re-reference the signal to the averaged signal from mastoid electrodes, due to excessive recording noise in the mastoid electrodes, we decided to proceed without re-referencing. This decision was taken based on data quality and prior to any further data analysis. Data were segmented into epochs starting from -1000 ms up to 2000 ms with respect to image onset time. Because the target images were presented on the screen for 2 seconds, and we allowed the participants to explore the scene visually, the EEG signal was largely contaminated by artefacts from horizontal and vertical eye movements. Therefore, an independent component analysis (ICA) using the FieldTrip toolbox [26] was carried out for artefact removal, since the rejection of the entire trials would have left too few trials for analysis. Components were visually inspected, and those corresponding to visual artefacts were removed. On average, two ICA components were removed per participant. All trials were visually inspected afterwards to assess the quality of the removal. As per the pre-registration document (https://osf.io/wftd7/), our focus was on Theta power (4–8 Hz) at mid-frontal location (Fz, Cz, FCz) in the time window between stimulus onset (0 ms) to 2000 ms post-stimulus. As a baseline, we considered the time window between -750 ms and -250 ms across all three stimuli types (incongruent, controls and fillers).

## Statistical analyses

**Theta power.** Theta power time course was calculated using a Fast Fourier Transform in 20 ms steps using a window of 500 ms and a Hanning tapper that included 3 cycles of the central frequency of the 4–8 Hz band of interest. Data were baseline corrected using the -750 ms to -250 ms period and transformed in dB. Then data were averaged across trials for the different conditions and each participant. Theta power was averaged across the electrodes of interest Fz, FCz and Cz and across the Theta band (4–8 Hz), obtaining a single time-course for each participant and condition.

We extracted the maximum Theta peak amplitude for each participant and condition in the time window between 50 ms and 600 ms post-stimulus from the average activity from electrodes Fz, FCz and Cz. This is the typical time window used for frontal-medial Theta analysis in the typical laboratory protocol [27–29] and it minimizes possible evoked effects due to image onset ($< 50$ ms). We then calculated the average amplitude within an 80 ms window centered on the peak (from -40 to +40 ms) for each participant and condition. Finally, we compared the peak amplitude between control and incongruent conditions separately, by

running a t-test (one-tailed; α = 0.05) with the hypothesis that peak amplitude was larger for incongruent than control images. An equivalent exploratory non-registered analysis was carried out on the latency of the peak (see below). We choose to initially test our hypothesis on the Theta peak (instead of the average activity though the entire time window) in order to preserve subject- and condition- peak differences in the Theta band [27, see for example 30].

**Memory test.**   For the behavioral memory test, we expected that incongruent images would be remembered better compared to control images [19]. Therefore, we tested whether the group-average correct response in incongruent trials was larger than control trials using a paired t-test (one tail; α = 0.05).

## Exploratory analyses

**Theta power.**   From the Theta power time course as described above, we also analyzed the latency of the Theta peak, comparing the peak latency between control and incongruent conditions separately by a t-test (two-tailed; α = 0.05). We also included a non-registered analysis that, instead of the single peak, compared the theta time-course from 250 ms to 1000 ms (time window decided in order to avoid potential contamination due to stimulus onset evoked activity). A right-tailed paired t-test was run with an alpha level of 0.05 for each time point within the window. Multiple comparison correction following Guthrie & Buchwald [31] determined a threshold of 7 consecutive significant (p< 0.05) points (φ ~ = 0.99, number of time points = 38, N = 26, and Theta = .05). The same interval was significant when correcting for the time window 0 to 1000 ms, which determines a threshold of 10 consecutive significant points. The same analysis was run but averaging across participants for each item to run the correlation by items.

**Alpha power.**   Beyond the pre-registered protocol, we also decided to explore the related hypothesis that conflict detection triggers a dynamic adjustment of cognitive control via attention re-orienting [32]. This hypothesis also is derived from the theoretical framework of cognitive control mechanisms [12]. According to this hypothesis, attention should be boosted right after incongruent images, compared to after controls. Similarly, as for Theta power, we calculated for each participant and trial the power between 8 and 14 Hz from the average activity from posterior electrodes (P3, Pz, P4, O1, O2). These electrodes where chosen given the expectation that attention effects would be best reflected in posterior scalp activity. We calculated the difference of the Alpha power between incongruent and control conditions for each participant. This difference was analyzed at group level by a right-tailed t-test (α = 0.05). As a signature of allocation of attention, we looked at the Alpha power (8–14 Hz) after image presentation. Indeed, a decrement of occipito-parietal Alpha power has been associated with effective attention allocation [32]. Similarly, as for the Theta power, we compared Alpha power after incongruent and control images in the time window 150–1000 ms in 20 ms steps, using a right-tailed t-test (α = 0.05) with the hypothesis that Alpha power would be lower for incongruent than for congruent conditions. However, the comparison turned out to be not significant after correction and will not be discussed any further.

**Memory test RTs.**   As for the pre-registered analysis regarding accuracy in the memory test, we also ran an analysis on the group average response times (RTs). Only correct response RTs within ± 2SD around the individual mean were considered for the analysis. As we did not have a hypothesis on RTs we applied a two-tailed t-test (p < 0.05).

**Item by item correlation between Theta power and memory performance.**   We explored the possibility that some relationship existed between the Theta power modulation found in the EEG analysis and the behavioural memory effect. Specifically, we tested whether images that produced a stronger Theta power response (raw Theta power averaged across participants

per image) were remembered better (i.e., higher accuracy and lower reaction times one week later) by an item-by-item correlation. To estimate Theta power for this correlation, we calculated the average raw Theta power in the window 370–710 ms, that had resulted significant at a group level when comparing the incongruent and control conditions. We performed the same analyses at a participant level, but in this case, instead of the raw Theta power, we calculated the average difference between incongruent and control images in the same time window as before, and then ran the correlation with memory performance.

## Results

As per the predictions from the pre-registered hypothesis, our focus was on the Theta power (4–8 Hz) at frontal-medial locations (reflected in Fz, Cz, FCz). We extracted the maximum Theta peak amplitude for each participant and condition in the time window between 50 ms and 600 ms after image presentation. The difference in the peak amplitude (average amplitude ± 40ms around the peak) in the Theta band between incongruent (1.49 ± 1.12 dB) and control (1.33 ± 1.33 dB) conditions was not significant [$t(25) = 0.69$; $p = 0.246$; $d_z = 0.135$]. However, the latency of the Theta peaks was different between conditions [incongruent 0.26 s ± 0.12; control 0.20 s ± 0.09; $t(25) = 2.09$; $p = 0.047$; $d_z = 0.409$], which we interpreted as a difference in time course of the frontal-medial Theta responses between conditions. We therefore decided to run a non-registered analysis on the frontal-medial Theta power, but across time. To do this, we compared frontal-medial Theta amplitude in response to incongruent and control images in the time window 250–1000 ms in 20 ms steps, using a right-tailed t-test ($\alpha = 0.05$) of the hypothesis that Theta power would be larger in incongruent compared to congruent conditions over time. Multiple comparison correction following Guthrie & Buchwald [31] determined a threshold of 7 consecutive significant ($p < 0.05$) points ($\varphi \sim = 0.99$, number of time points = 38, N = 27, and Theta = .05). We found significant differences between incongruent and control images in frontal-medial Theta power between 370 ms and 710 ms, with 18 consecutive significant points, revealing that looking at incongruent images was associated with a higher Theta power compared to controls (see Fig 2). It is important to note that the two analyses (the original pre-registered analysis on the Theta peak and the Theta power time evolution) address different aspects of the Theta response, but both involve the same (pre-registered) hypothesis that incongruent images would elicit stronger Theta power compared to controls in the time window after image presentation.

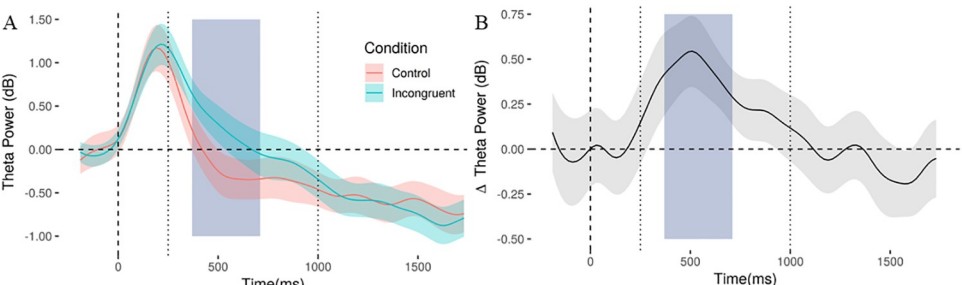

**Fig 2. Theta power (4–8 Hz) calculated from mid-frontal electrodes (Fz, Cz, FCz) as a function time (0 = image onset) for (A) the incongruent and control conditions separately, and (B) as the difference between the two conditions.** The dotted vertical lines indicate the time window used for the analysis (250–1000 ms), while the grey area show the interval (370–710 ms) in which the two conditions significantly differ after multiple comparison correction. The shaded areas around the lines represent the standard error of the mean.

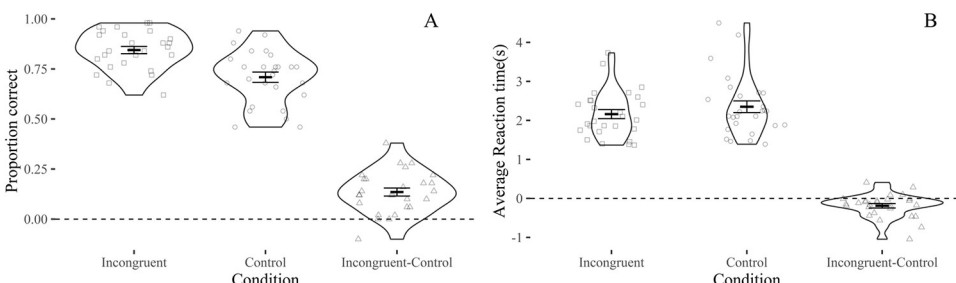

**Fig 3. Results from the memory test in terms of accuracy (A) and reaction times (B) for incongruent and control images, and as the difference between the two conditions.** Each dot represents a participant, bars show the group average and the standard error of the mean.

As we were also interested in the memorability of the incongruent images compared to controls, we invited participants to come back to the lab one week after the EEG recordings and perform a memory test: they were asked to choose which of two images presented side-by-side they remembered seeing during the EEG session (two-alternative forced-choice -2AFC- task) (Fig 1B). As expected, incongruent images were remembered more often (hit = 84% ± 9%) compared to control images (hit = 71% ± 13%) [t(26) = 6.77; p = $3e^{-07}$, $d_z$ = 1.3]. Mean reaction time (RT) for incongruent images (2.2 s ± 0.60) was faster than for control images (2.4 ± 0.8) [t (26) = -3.28; p = 0.00291, $d_z$ = -0.63] (Fig 3).

Given the positive results on both the Theta power modulation and the memory test, we decided to run an item-by-item correlation between the two measurements (exploratory analysis). We calculated the item-by-item accuracy as the proportion of correct responses across participants, and the item-by-item Theta power by averaging it across participants. We would like to highlight that not all images were presented to all participants (see Experiment 1), but they were counter-balanced across them. We found a small but significant correlation at item-by-item level between Theta power and memory accuracy (t = 2.180, df = 198, p-value = 0.030, 95% CI = [0.015 0.29], r = 0.15). Our data support that the effect is positive as the CI excludes the r = 0 value, albeit it may vary from a very modest size effect from 0.015 to 0.29. This correlation suggests that the images that evoked higher frontal-medial Theta power responses are the ones more often remembered one week after the EEG phase (Fig 4).

At a participant level, we found no correlation between the Theta power difference and memory accuracy (t = -1.888, df = 25, p-value = 0.070, 95% CI = [-0.65 0.03], r = -0.35). This means that participants with stronger Theta modulation are not necessarily the ones who will remember images more successfully. The correlation between Theta power and the reaction times was not significant neither across items (t = -0.122, df = 198, p-value = 0.903, 95% CI = [-0.15 0.13], r = -0.009) or participants (t = -1.45, df = 25, p-value = 0.158, 95% CI = [-0.60 0.11], r = -0.28).

## Discussion

We tested the hypothesis that images containing incongruences, such as surrealistic images often used in advertising, trigger cognitive conflict responses in the brain. We also tested the related assumption that episodic memory for images with incongruent content would be superior to other, congruent images. We based our predictions on the conflict monitoring theory [3,12–15,33], which allowed us to be specific regarding the expected results (see pre-registration https://osf.io/wftd7/). We measured frontal-medial Theta EEG activity, a reliable

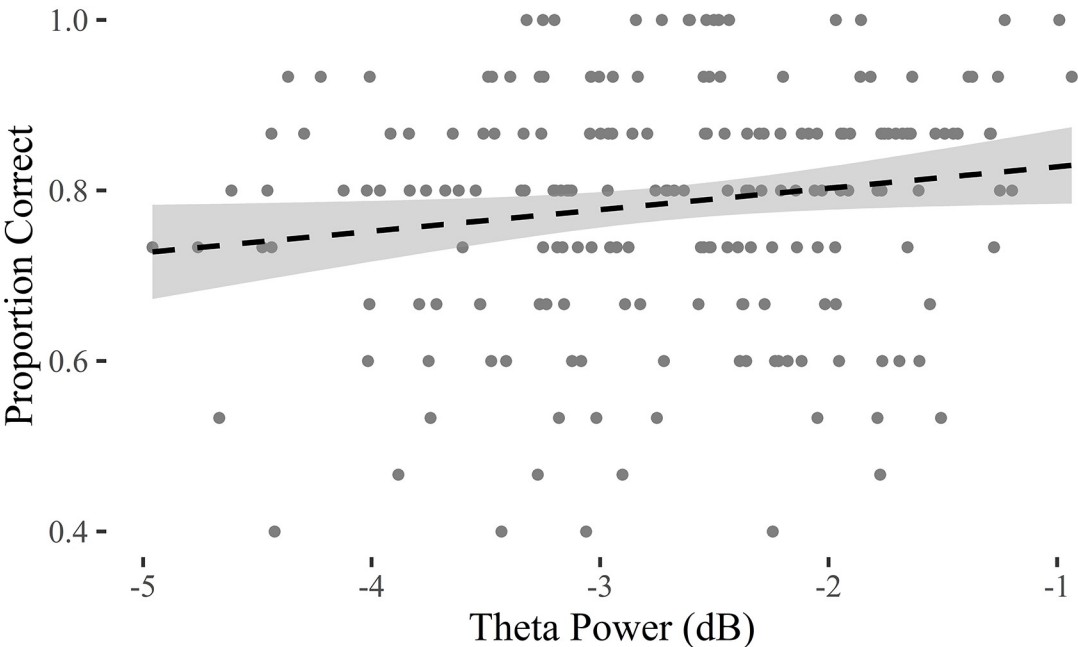

**Fig 4. Image-by-image correlation between induced Theta power (x-axis) and the proportion of correct responses (y-axis) at the memory test.** Least squares linear fit and its confidence interval are shown with the dashed line and shaded area, respectively.

neuromarker of cognitive conflict [10], following the presentation of incongruent images selected from media advertising compared to visually similar but realistic images. As predicted, albeit with a non-registered analysis pipeline, we found that incongruent images not only generated stronger Theta activity at frontal-medial scalp locations compared to control images, they also produced higher recognition rates in a memory task delivered one week after the exposure. Considering that we specifically selected incongruent images mostly from real ads designed according to surrealistic techniques, higher memorability for those images can be seen as a key behavioral metric of advertising effectiveness [34]. Furthermore, item-by-item Theta power and subsequent memory recognition for the corresponding image showed a weak, albeit significant correlation. This suggests that the stronger the Theta response an image elicits in the brain, the more likely it will be remembered.

Incongruent images generate a pattern of brain activation (i.e., frontal-medial Theta power increment) similar to the one seen in the conflict studies that have grounded cognitive control theories [3,13]. Based on this, we suggest that surrealistic images using the juxtaposition of incongruent elements generate conflict at a cognitive level and can be accounted for by these theories.

Consistent with our account, Mostafa [16] used fMRI to measure BOLD responses during passive viewing of advertisements, exploring which brain areas were activated by surrealistic images. He found that surrealistic images elicited greater activation in the ACC (among several other areas), compared to other images. Mostafa also suggested that the violation of expectations induced by surrealistic images may favor later recall, although he did not test memory recall explicitly. Here, we were able to confirm this assumption: surrealistic images are remembered better than controls. This hypothesis is implicit in real-world advertising, although previous cognitive neuroscience findings show that, indeed, incongruent items in a scene are

memorable. For example, Krebs and collaborators [19], explored the impact of cognitive conflict on memory using a modified version of the Stroop task (i.e., the face-word Stroop task), showing that faces associated with incongruent stimuli were remembered more. In another study, by Friedman [18], participants noticed only the changes that had been made to objects that were unexpected based on the surrounding scene (e.g., a coffeemaker shown in a farmyard, compared to a kitchen). Finally, Homer & Kahle [2] found that participants who were exposed to surrealistic ads were better at free-recall, compared to a control group exposed to non-surrealistic ads. In line with the mentioned studies, the present study showed that incongruent images were remembered better than controls one week later. Remarkably, frontalmedial Theta responses to images correlated with memory performance on image-by-image basis. Although the effect size of this correlation is small, it suggests a potential link between brain mechanisms of cognitive control and encoding into episodic memory. This link can find application in the context of advertising strategies and help ground a long-standing assumption in advertising through surrealistic images: images containing the juxtaposition of incongruent elements are memorable. Our results provide a viable experimental procedure and proof-of-concept data that helps bridge the gap between a scientific framework based on cognitive neuroscience and advertising, uncovering new insights into the relationship between cognitive conflict and incongruent images such as surrealistic ads.

The type of conflict addressed in the present study is generated by incongruence among visual properties of a stimulus, independently of conflict at the response level. The original version of the conflict monitoring theory proposed that conflict could arise at various stages of information processing, from perceptual representation to stimulus categorization and response selection [12]. Yet, to date, most of the studies related to frontal-medial Theta power and cognitive control struggle to disambiguate between response conflict and stimulus conflict [33]. Given the nature of our paradigm, pure stimulus conflict was relevant, and we only analyzed the response from image onset, avoiding response preparation or motor activity using a delayed response protocol and random assignment of response sides. Our results suggest that taking a step away from typical laboratory tasks can lead to discoveries that are of interest to the theory itself, in addition to real-life applications, although the experimental control on image selection has to be considered as a potential limitation because it is a source of variability.

The idea (and the evidence provided here) that cognitive control mechanisms might go in line with the processing of surrealistic images establishes a fruitful connection between neurocognitive theories and advertising. For example, although it is often assumed that humans avoid conflict because it is energetically costly (physical and/or mental effort) [35,36], one suggestive, and speculative, possibility is that conflict resolution could also act as an internal reward [15,37]. Recently, Inzlicht et al. [15] proposed a computational model in which effort can be considered both a cost and an added value which works as intrinsic reward and motivation. Incongruent images may require extra processing (i.e., effort) to be interpreted, which could have positive consequences (i.e. stronger memory). Therefore, extending the proposal by Inzlicht, et al. [15], the surrealistic strategy used in advertising and arts not only elicits cognitive conflict and increased memory, as our data suggest but might also be seen as an intrinsic reward, potentially linked to positive subjective experience [38,39]. However, this remains a question to be tested in future studies.

So far, we have only considered the interpretation that surrealistic advertising images elicit cognitive conflict; but, alternatives should be explored. First, in this study, we have operationalized our hypothesis capitalizing on semantic incongruence, which is a typical strategy in surrealistic techniques. However, one should consider that surrealistic images are not solely (or even necessarily) defined by incongruent contents. Therefore, our conclusions are limited to

these conflict situations. Second, the Theta power enhancement at frontal-medial locations is considered a reliable measure of cognitive conflict, as demonstrated by the classical experimental paradigms (i.e., Stroop, Flanker, Simon tasks) of cognitive control [7,29]. However, Theta power enhancement has also been associated with novelty, errors, or even feedback on errors [29]. Because the task during the EEG recordings was orthogonal to the congruency manipulation, it is doubtful that the Theta effects could be attributed to participant errors or negative feedback. In fact, no errors could be possible, nor feedback was provided. In addition, an element of increased novelty is necessarily conflated with surrealism, and therefore its contribution to our main finding should be considered. Indeed, we decided to select the incongruent images based on the *originality* and *synthesis* dimensions of ad divergence as defined by Smith et al. [21], which both imply novelty to some extent.

In conclusion, this study provides an original piece of evidence regarding the potential of surrealistic images to generate cognitive conflict in the brain and to create memorable impressions, as was the original spirit of the surrealist movement at its birth: *"The image is a pure creation of the mind. It cannot be born from a comparison but from a juxtaposition of two more or less distant realities. The more the relationship between the two juxtaposed realities is distant and true, the stronger the image will be—the greater its emotional power and poetic reality. . ."* *Pierre Reverdy, 1918 in Manifesto of Surrealism* [40].

## Acknowledgments

We wish to thank Dr. Marc Lluís Vives for his comments to a preliminary version of the manuscript and the discussions about the topic.

## Author Contributions

**Conceptualization:** Manuela Ruzzoli, Aoife McGuinness, Salvador Soto-Faraco.

**Data curation:** Manuela Ruzzoli, Aoife McGuinness.

**Formal analysis:** Luis Morís Fernández.

**Funding acquisition:** Manuela Ruzzoli, Salvador Soto-Faraco.

**Investigation:** Manuela Ruzzoli, Aoife McGuinness.

**Methodology:** Manuela Ruzzoli, Aoife McGuinness, Salvador Soto-Faraco.

**Software:** Luis Morís Fernández.

**Supervision:** Manuela Ruzzoli, Luis Morís Fernández, Salvador Soto-Faraco.

**Writing – original draft:** Manuela Ruzzoli, Aoife McGuinness.

**Writing – review & editing:** Manuela Ruzzoli, Aoife McGuinness, Luis Morís Fernández, Salvador Soto-Faraco.

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
