## [Decision Letter · Decision Letter 0]

20 Jan 2020

PONE-D-19-27606

From cognitive control to creativity: The conflict resolution of surrealistic imagery

PLOS ONE

Dear Dr. Ruzzoli,

Thank you for submitting your manuscript to PLOS ONE. After careful consideration, we feel that it has merit but does not fully meet PLOS ONE’s publication criteria as it currently stands. Therefore, we invite you to submit a fully revised version of the manuscript that addresses all the points raised during the review process.

I require that the revised manuscript method have to be amended according to the Reviewer 1 requirements and the results have to include the analyses required by Reviewer 2.

We would appreciate receiving your revised manuscript by Mar 02 2020 11:59PM. To enhance the reproducibility of your results, we recommend that if applicable you deposit your laboratory protocols in protocols.io, where a protocol can be assigned its own identifier (DOI) such that it can be cited independently in the future. For instructions see: http://journals.plos.org/plosone/s/submission-guidelines#loc-laboratory-protocols

We look forward to receiving your revised manuscript.

Kind regards,

Francesco Di Russo, Ph.D.

Academic Editor

PLOS ONE

Journal Requirements:

2. Our internal editors have looked over your manuscript and determined that it may be within the scope of our Neuroscience of Reward and Decision Making Call for Papers. This collection of papers is headed by a team of Guest Editors for PLOS ONE: Stephanie Groman, Satoshi Ikemoto, Jane Taylor and Robert Whelan. With this Collection we hope to bring together researchers working on a wide range of disciplines, from animal subjects research, computational approaches and patient-centered research. Additional information can be found on our announcement page: https://collections.plos.org/s/reward-and-decision-making. If you would like your manuscript to be considered for this collection, please let us know in your cover letter and we will ensure that your paper is treated as if you were responding to this call. Agreeing to be part of the call-for-papers will not affect the date your manuscript is published. If you would prefer to remove your manuscript from collection consideration, please specify this in the cover letter.

3. Please modify the title to ensure that it is meeting PLOS’ guidelines (https://journals.plos.org/plosone/s/submission-guidelines#loc-title). In particular, the title should be "specific, descriptive, concise, and comprehensible to readers outside the field" and in this case  it is not informative and specific about your study's scope and methodology.

Reviewers' comments:

Reviewer's Responses to Questions

**Comments to the Author**

1. Is the manuscript technically sound, and do the data support the conclusions?

Reviewer #1: No

Reviewer #2: Yes

2. Has the statistical analysis been performed appropriately and rigorously? 

Reviewer #1: Yes

Reviewer #2: Yes

3. Have the authors made all data underlying the findings in their manuscript fully available?

Reviewer #1: Yes

Reviewer #2: Yes

4. Is the manuscript presented in an intelligible fashion and written in standard English?

Reviewer #1: No

Reviewer #2: Yes

5. Review Comments to the Author

Reviewer #1: The paper describes a potentially sound research question, exploring the brain correlates and the memory processing associated to the vision of incongruous images in the advertisement domain. Even if this research question could be particularly intriguing, the work has too many theoretical and methodological flaws to be considered publishable in Plos One in my opinion. In the following, major and minor criticisms to the paper are listed:

INTRODUCTION

- The introduction is only drafted:while the aim of the paper can be grasped, neither a theoretical proposition, not specific hypotheses have been described by the authors. The preregistration of the work is a valuable add to a paper, but the published paper must be a self-standing work; it is indeed necessary that the text contains all information necessary to the reader to understand the theoretical propositions, the aims, the originality, and the hypotheses of the work. At the moment, none of these points is adequately described in the introduction. Moreover, while the pre-registered hypotheses were simple and clear, in the present paper the theoretical and methodological questions seem to shift from incongruity to surrealism, which are distinct concepts. The research questions seem therefore to be different from the questions described in the pre-registered work.

- The authors wrote "In the present study, we tested the hypothesis that visual imagery that contains incongruent meanings, such as that used in advertising,..." This sentence is not clear, representing a generalization of the use of visual imagery in advertising.

- The authors wrote: "Conflict response will be reflected in frontal-medial Theta power (4-8Hz), compared to control images that do not contain incongruity." Please reformulate: the two terms of the contrast are posed on different levels (conflict response vs control images)

- The authors wrote: "Furthermore, we assess the impact of conflict on subsequent memory by introducing a recall task one week after exposure to the images. We address the specific hypothesis regarding the memorability of surrealist imagery, implicit in advertising. Cognitive neuroscience has suggested a relationship between visual memory capacity and semantic consistency (see Brady, Konkle, & Alvarez, 2011), whereby objects that are inconsistent with the surrounding scene are remembered more (see also Friedman, 1979; Krebs, Boehler, De Belder, & Egner, 2015). The impact of surrealistic ads on memory has been only tested in a seminal study by Homer & Kahle (1986) using a free-recall test. They found that participants who saw surrealistic ads were less inclined to errors, compared to a control group, thus supporting the proposal that surrealistic images strengthen memory." Please be more precise and specific:

- “ after exposure to the images” which kind of images?

- imagery or images? Surrealist imagery and surrealist images are two different concepts.

- Please be more specific on the main features and distinctive elements defining surrealistic ads. This is a critical point of the paper, which never defines the main constituents of surrealistic images, surrealistic imagery, and surrealistic ads (which are three different concepts and research fields). Moreover, a raw generalisation from surrealism to incongruity has been made by the authors, which is a high level simplification of the concept of surrealism.

- The authors defines conflict as a creative device; however, it is hard to define “conflict” as a creative device, whereas it is usually used as a creative technique or as an ideative method.

METHOD

- The method used to operationalize the construct of incongruity is essentially based on wrong assumptions, since it used a methodology to define and measure a different construct: divergence.

- The authors wrote: "All 240 images initially selected with the methods above were then subjected to a survey to validate the levels of visual incongruity for each image by independent judges. The survey consisted of determining the degree of agreement of two statements regarding the categories of originality and synthesis from Smith et al.’ study (Smith et al., 2007), which can be considered an objective way to determine image incongruity. " The two categories used by Smith et al. (2007) are only two of a series of categories defining the concept of divergence (applied to the specific case of advertising). All these categories were extracted from classical definition of divergent thinking (using the criteria by Guilford, Torrance, etc.). Incongruity and divergence (especially if referred to divergent thinking) are, from a theoretical, practical, and methodological points of view, distinct and different psychological constructs. The fact that they can overlap requires a specific scientific analysis, which is not performed in the present work. All in all, the choice of a method used to measure a construct to define a different construct is scientifically and technically wrong.

- Moreover, since perception represents a critical key in the processing of congruency/uncongruency images must be controlled for their physical properties (at least high level processes).

RESULTS

- Some imprecisions in reporting and interpreting the findings are present in the results section, which then engender a series of misunderstanding and wrong assumptions in the final discussion.

- The authors wrote: "Multiple comparison correction following Guthrie & Buchwald (1991) determined a threshold of 7 consecutive significant (p< 0.05) points (φ ~= 0.99, number of time points = 38, N=27, and Theta = .05). We found significant differences in frontal-medial Theta power in a 340 ms time window (between 370 and 710 ms), with 18 consecutive significant points, revealing that looking at incongruent images was associated to a higher Theta power compared to controls (see Figure 2)." It is here highly important to introduce and comment the meaning of this result. The difference is not an overall difference, but it is related to the time (and expecially to a specific time window). This is a fundamental distinguo, which is at the core of the emerging results.

- Correlations must be read/interpreted/discussed on the basis of the effect size and not on the basis of the statistical significance. Looking a the correlations on items and on individuals, the size of the effect is almost twice in the latter case (and negative) than in the former case.

DISCUSSION

- Starting from the first sentences of the discussion section it is still not clear the distinction between incongruent and surrealistic images. Are they the same? Why? Why not defining the images simply as incongruent as made in the registered protocol?

- The authors wrote: "As predicted, albeit with a non-registered analysis pipeline, we found that surrealistic advertising imagery not only generated stronger Theta activity at frontal-medial scalp locations compared to control images, it also produced higher recognition rates in a memory task delivered one week after the exposure, a key behavioral metric of advertising effectiveness (Keller, 1987). Furthermore, item-by-item Theta power and subsequent memory recognition for the corresponding image showed a correlation. This suggests that the stronger the Theta response an image elicits in the brain, the more likely it will be remembered, albeit the effect size of this correlation was small." Results must be reported as they are. Authors find a difference in Theta only in relation to temporal dynamics and not as an overall effect; this point is critical and must be discussed. Moreover, the correlations showed that the variables are really weakly correlated, which is exactly the contrary of what authors expected and are sustaining (in the case of individual correlations the variables are negatively associated - with a stronger effect size than the effect discussed by the authors).

- The authors wrote: "Our results provide a viable experimental procedure and proof-of-concept data that helps bridge the gap between the scientific and creative approaches and uncover new insights into the relationship between cognitive conflict and creativity." It is totally unclear why it should give insight on this relationship. Creativity has not been explored in the present study.

- The authors wrote: "Therefore, extending the proposal by Inzlicht, et al. (2017), the creative communication strategy typically used in advertisement and arts not only elicits cognitive conflict and increased memory, as our data suggest, but might also be seen as an intrinsic reward, potentially linked to positive subjective experience (Chetverikov & Kristjánsson, 2016; Van de Cruys & Wagemans, 2011)." This is pure speculation which can not directly derived from the present results; ad hoc experimental studies should be made based on these hypotheses. What do you mean for “creative communication strategy”? The (logical, conceptual, operational) gap between the images used in this study and a typical creative communication strategy is, to say the last, huge.

Reviewer #2: By means of surrealistic images (semantically conflicting) presentation, this study has two main aims. The first one is to confirm the increase in theta oscillatory activity (4-8 Hz) over mid-frontal cortex; the second one is demonstrate, for the firs time, the neural basis of the increase memorability of the surrealistic semantic conflicting images. The results confirm the two hypothesis showing that observing incongruent images was associated to a higher theta power compared to controls and that viewing incongruent images also produced higher recognition rates compared to control images. The study is performed accurately and the results are interesting. However, in order to improve the quality of the paper and to deeper understand the results, I suggest to perform a new analysis comparing theta and alpha power in the recording sites.

Major points

My first methodological concern is about the category of incongruent images that is a bit vast and, for what I understood, it comprising a mix of images including art, advertising, objects, human and non human action, etc. Also the control images seem to be rather mixed. A categorization of the images belonging to control and experimental conditions and a precise matching between each category in both conditions should be given.

The survey performed in order to measure the degree of congruency/incongruency has two questions but it is not explained which one has been used for the images selection. Please, provide this information together with the two average scores (one for each question) of the 100 incongruent and 100 congruent images used as final experimental image set. Ideally the same survey should be done also on the filler images in order to exclude a bias of these images for congruency or incongruency scores.

About the Theta and Alpha power I strongly suggest to perform a 2X2X2 Anova with the main factor of Condition (congruent and incongruent), Rhythm (theta and alpha) and Location (fronto-central cluster and parietal-occipital cluster). This analysis allows having more realistic overview of the brain responses during the observation of the two categories of images. In particular, it could allow to demonstrate that the increase of theta power is specific for fronto-central brain region.

Please better describe the item by item correlation analyses.

Figure 2 is incomplete: in the time scale please replace s with ms, add the time values of the window used for the analyses and the time values of the window that resulted significant in the comparison between the two conditions. Add a plot showing the theta power for control condition

Fig 4: what does proportion of correct responses (y-axis) mean, how it is calculated?

6. PLOS authors have the option to publish the peer review history of their article (what does this mean?). If published, this will include your full peer review and any attached files.

Reviewer #1: No

Reviewer #2: No

---

## [Decision Letter · Decision Letter 1]

11 May 2020

From cognitive control to visual incongruity: Conflict detection in surrealistic images

PONE-D-19-27606R1

Dear Dr. Ruzzoli,

We are pleased to inform you that your manuscript has been judged scientifically suitable for publication and will be formally accepted for publication once it complies with all outstanding technical requirements.

With kind regards,

Francesco Di Russo, Ph.D.

Academic Editor

PLOS ONE

Additional Editor Comments (optional):

Reviewers' comments:

Reviewer's Responses to Questions

**Comments to the Author**

1. If the authors have adequately addressed your comments raised in a previous round of review and you feel that this manuscript is now acceptable for publication, you may indicate that here to bypass the “Comments to the Author” section, enter your conflict of interest statement in the “Confidential to Editor” section, and submit your "Accept" recommendation.

Reviewer #2: All comments have been addressed

2. Is the manuscript technically sound, and do the data support the conclusions?

Reviewer #2: Yes

3. Has the statistical analysis been performed appropriately and rigorously? 

Reviewer #2: Yes

4. Have the authors made all data underlying the findings in their manuscript fully available?

Reviewer #2: Yes

5. Is the manuscript presented in an intelligible fashion and written in standard English?

Reviewer #2: Yes

6. Review Comments to the Author

Reviewer #2: All my points have been addressed and I consider the manuscript now suitable for publication in PlosOne.

7. PLOS authors have the option to publish the peer review history of their article (what does this mean?). If published, this will include your full peer review and any attached files.

Reviewer #2: No

---

## [Editor Report · Acceptance letter]

26 May 2020

PONE-D-19-27606R1 

From cognitive control to visual incongruity: Conflict detection in surrealistic images 

Dear Dr. Ruzzoli:

I am pleased to inform you that your manuscript has been deemed suitable for publication in PLOS ONE. Congratulations! Your manuscript is now with our production department. 

With kind regards,

on behalf of

Prof. Francesco Di Russo 

Academic Editor

PLOS ONE